# Synergism of Photo-Induced Electron Transfer and Aggregation-Induced Quenching Mechanisms for Highly Sensitive Detection of Silver Ion and Captopril

**DOI:** 10.3390/molecules28093650

**Published:** 2023-04-22

**Authors:** Jing Zhu, Lei Hu, Xiangying Meng, Feng Li, Wenjuan Wang, Guiyang Shi, Zhongxia Wang

**Affiliations:** 1School of Chemistry and Chemical Engineering, Yancheng Institute of Technology, Yancheng 224051, China; 2School of Medical Laboratory, Weifang Medical University, Weifang 261053, China

**Keywords:** carbon-based nanoprobe, photo-induced electron transfer, aggregation-induced quenching, metal ion detection, drug analysis

## Abstract

Carbon-based nanoprobes, with excellent physicochemical performance and biocompatibility, are a kind of ideal nanomaterial for biosensing. Herein, we designed and prepared novel oxygen-doped nitrogen-enrichment carbon nanoribbons (ONCNs) with an excellent optical performance and uniform morphology, which could be used as a dual-mode fluorescence probe for the detection of Ag^+^ ion and captopril (Ctl) based on the synergism of photo-induced electron transfer and aggregation-induced quenching mechanisms. By recording the changes in fluorescent intensities of ONCNs, the Ag^+^ ion and Ctl concentrations can be easily tested in real samples. The results displayed that two good linear relationships existed between the change in fluorescent intensity of ONCNs and the concentrations of Ag^+^ ion and Ctl in the ranges of 3 μM to 30 μM and 1 μM to 30 μM, with the detection limit of 0.78 µM and 74 nM, respectively. The proposed sensing platform has also been successfully applied for the Ctl analysis in commercial tablet samples based on its high selectivity, proving its value in practical applications.

## 1. Introduction

Fluorescent carbon-based nanomaterials (FCNMs), as a revolutionary luminescent material, have attracted intense research interests in many research fields, with applications ranging from biosensing to bioimaging, drug delivery, optoelectronics, etc. [1,2,3,4]. Compared with organic probes and traditional semiconductor quantum dots, FCNMs are attractive mainly because of their strong fluorescent emission, facile preparation, low cost, high chemical stability, environmental friendliness, and especially excellent biocompatibility [4,5]. Recently, researchers have found that the internal structure and electron distribution of FCNMs can be altered through doping different heteroatoms during the synthesis process. Among the numerous doped elements such as nitrogen, boron, and sulfur elements, nitrogen doping is popular for its comparable atomic size and five valence electrons that can be used to bond with carbon atoms. For example, Li et al. reported nitrogen-doped carbon dots for the selective determination of Ag^+^ using salecan and dicyandiamide as the precursors [6]. Gu et al. prepared nitrogen-rich carbon nanodots by a simple solvothermal method for the specific determination of cysteine [7]. Zhang et al. reported a core-shell nitrogen and boron codoped carbon dots with excellent water solubility, remarkable stability, and a high fluorescence quantum yield [8]. Yang et al. presented red-emitting carbon dots for the specific monitoring polarity of solutions in living cells [9]. Despite these good examples, continuous efforts are still needed to fabricate doped FCNMs for a wide variety of applications.

Captopril (Ctl), as one of the angiotensin-converting enzyme (ACE) inhibitors, is widely used in the treatment of hypertension, congestive heart failure, myocardial infarction, and kidney problems caused by diabetes [10]. Generally, administering Ctl for therapeutic use can significantly reduce cardiovascular morbidity and mortality. Toxicity from Ctl is uncommon, but when Ctl is given in high doses to patients, it can lead to undesirable side effects, such as kidney damage, proteinuria, bone marrow suppression, circulatory failure, and other symptoms [11]. Thus, it is essential to closely monitor Ctl levels in pharmaceutical samples to guide the clinical use of drugs. As far as the literature survey, the primary methods include high-performance liquid chromatography, capillary electrophoresis, electroanalytical methods, chemiluminescence, and fluorescent method for the determination of Ctl [11,12,13]. Among these methods, the fluorescent method has attracted more and more attention in recent years because of its high sensitivity, simple operation, less expensive cost, and real-time monitoring. Therefore, the efficient analysis technique based on the fluorescent method for Ctl detection is still a hot topic and a huge challenge. Moreover, silver is one of the representative noble metal elements, which is widely used in various fields, such as chemical engineering, pharmaceutical preparations, cosmetics, and dressings [14]. Unfortunately, the excess of Ag^+^ ion in the human body can severely disrupt biological systems, resulting in growth retardation, mitochondria damage, inhibition of organ function, etc. [15,16,17]. Considering that the conventionally used methods, such as atomic absorption spectrometry, atomic emission spectrometry, and inductively coupled plasma mass spectrometry, are expensive, lack convenience, and involve complex operating procedures [18], it is highly essential to develop a simple, sensitive, and convenient analytical method for the accurate detection of Ag^+^ ion from a clinical point of view.

Herein, we successfully synthesized water-soluble oxygen-doped nitrogen-enrichment carbon nanoribbons (ONCNs) by simple hydrothermal method and used them as a fluorescent probe for the highly sensitive and selective detection of Ag^+^ ions and Ctl through synergism of photo-induced electron transfer and aggregation-induced quenching mechanisms. The results display an interesting “on-off-on” three-state emission with the stepwise addition of Ag^+^ ions and Ctl (Figure 1). Firstly, the fluorescent intensity of ONCNs decreases in the presence of Ag^+^ ions due to the great affinity of Ag^+^ towards the N to form the non-fluorescent complex. Further, with the addition of Ctl into the non-fluorescent complex, the quenched ONCNs fluorescence is restored because of the existence of the thiol group on Ctl favors the formation of the Ag-S bond, leading to the detachment of Ag^+^ ions from ONCNs. Based on the excellent analytical performance of this sensing platform, such as good sensitivity and selectivity, wide linear range, and acceptable accuracy, it is considered to have great prospects in the detection of Ag^+^ ions and Ctl.

## 2. Results and Discussion

The synthesis steps are shown in Materials and Methods. PLA and DMF were used as the carbon–oxygen–nitrogen source and stabilizer, respectively, without adding any other toxic reagents, making the whole synthesis process green and environmentally friendly. Morphology of the prepared fluorescent ONCNs were characterized by TEM, as shown in Figure 1A. It is shown that ONCNs are nearly uniform-shaped and have a needle-like shape, and all ONCNs are almost monodisperse and less than 250 nm in length-diameter (Figure 1A), which is further clearly seen by enlarged TEM image (Appendix A).

In order to reveal the state of O- and N-doping in ONCNs, the FT-IR and XPS spectra were collected from dried solid ONCNs samples. As shown in Appendix A, the chemical formation of ONCNs was studied by FT-IR spectrum, and the results display that N mainly exists as an organo-amine group (N-H stretched at 3054–3388, bent at 1451, and waged at 762 cm^−1^) [19,20]. Meanwhile, there were a lot of other active groups on the ONCNs’ surface. The three strong peaks at 1702, 1626, and 1401 cm^−1^ correspond to the stretching vibrations of the C=O bond, covalent C=N, and C-N, suggesting the sp^2^-hybridized carbon structure of the carbon core [9,19,21]. The asymmetric and symmetric stretching vibrations of the C-O bond at 1246 and 1087 cm^−1^ [22], respectively, and the stretching vibration of the C-O-C bond at 1147 cm^−1^ [23]. The stretching and bending peaks of methylene (-CH_2_ groups) from 2918 and 973 cm^−1^ underwent some changes that appeared relatively [24]. Moreover, the results of XPS proved that the as-prepared ONCNs were composed of three elements, which were C, N, and O, corresponding to C 1s (286.8 eV), N 1s (402.8 eV), and O 1s (531.8 eV) [25], as displayed in Figure 2A. The elemental analysis result of ONCNs shows a predominant presence of the C (74.69%) and N (16.56%) elements in the FL ONCNs, whereas the amount of O elements was limited (8.80%) (inset of Figure 2A). The high-resolution XPS spectrum of C1s (Figure 2B) peaking at 284.6, 285.2, 286.1, and 288.7 eV correspond to C-C/C=C, C=N, C=O, and O-C=O, respectively [21,24]. The N1s spectrum peaking at 399.5, 400.2, 401.1, and 401.8 eV ascribe to pyridinic-N/pyrrolic-N, graphitic nitrogen, and amino-N (401.1 eV for primary amino-N, and 401.8 eV for secondary amino-N) [26,27], respectively (Figure 2C). The high-resolution O1s spectrum peaking at 531.1, 531.9, and 532.9 eV assign for C=O, C-OH, and O-C=O, respectively (Figure 2D) [22,28]. Furthermore, the synthesized ONCNs can be well dissolved in water solution because of the hydrophilic groups such as C=NH, -NH_2_, -OH, and O=C-OH groups on the ONCNs surface, which makes the prepared ONCNs possess excellent physical and chemical properties with good biocompatibility.

Next, the FL and UV-vis spectroscopies were carried out to examine the optical properties of ONCNs. As displayed in Figure 3A, the UV-vis spectrum of the as-prepared ONCNs shows two typical peaks at ~256 and ~335 nm (black curve, Figure 3A). The former peak at ~256 nm corresponds to the σ → π transition of the aromatic sp^2^ domains [26], which cannot generate FL signal. The latter peak located at ~335 nm belongs to the n → π* electron transition of the C=N/C=O groups in the skeleton of ONCNs [29,30], which can generate strong FL emission at 390 nm due to the trapping of the large extent of excitation energy of C=N/C=O groups (Figure 3A, red-dot and red lines). Meanwhile, to study the FL behavior of the ONCNs under different excitation wavelengths, the excitation-dependent FL behavior was investigated, as shown in Figure 3B. Like most of the reported FL carbon-based nanoprobes [22,31,32,33], the emission peak position of the ONCNs did not change when the excitation wavelength varied from 315 nm to 355 nm, exhibiting an excitation-independent FL behavior. This phenomenon might be owing to the formation of uniform radiating surface recombination channels by efficiently trapping the excited electrons of the C=N/C=O transition of the surface state.

In order to evaluate the application value of the as-prepared ONCNs, the FL stability of the prepared ONCNs under different conditions such as solution pH, ionic strength, and storage time was tested in detail. As shown in Appendix A, these results were obtained through experimental tests: (a) the ONCNs have good stability under different ionic strength conditions (used NaCl to control ionic strength), even up to 0.3 M NaCl (Appendix A); (b) the photostability of the ONCNs is relatively stable in a wide range of pH value 3–12 (Appendix A); (c) the FL stability of ONCNs is very good in a long storage time (Appendix A). Note that when the pH value was higher than 12 or lower than 3, the FL intensities of ONCNs gradually increased or decreased, respectively. The main reason might be that there were many retained nitrogen-containing (e.g., -NH_2_ or C=NH) and oxygen-containing groups on the surface/edge of ONCNs, which could exist in different states in high (-NH_2_) or low pH (-NH_3_^+^) environments, resulting in structural rigidity or flexibility through hydrogen bonding, ultimately leading to fluorescence enhancing or quenching by OH^−^ or H^+^. These results demonstrate that the ONCNs have excellent stability, which might be attributed to the electrostatic repulsion between charged nanoribbons.

As a strong Lewis acid, Ag^+^ ion (4d^10^5s^0^) has a strong affinity with the Lewis base of the nitrogen-donor groups in the probe, resulting in the possible formation of the stable metal complexes between Ag^+^ ion and nitrogen-donor of the surface of the probe through strong Ag^+^-N bonds [34] and leading to formed nonfluorescent or FL-enhancement metal chelate through the electron/energy transfer process. In addition, compared with the solubility product constant (K_sp_) between Ag^+^ and a nitrogen atom, Ag(I) can interact strongly with mercapto groups via coordination/chelation due to the lower K_sp_ between Ag^+^ and -SH [22], which could lead to a significant recovery of FL after the detachment of Ag^+^ ions from ONCNs. On that basis, the feasibility of the new “on-off-on” nitrogen-containing ONCNs probe was evaluated by adding Ag^+^ ions and the mercapto-containing Ctl drug. In light of our previous work and other relevant research, the changes of absorption peak (∼335 nm) and emission peak (~390 nm) could be caused by the difference in the state of nitrogen-containing groups (C=N, -CONH-, or -NH_2_) under different conditions (Figure 3A) [34]. That is, the binding of ONCNs with Ag^+^ ions caused a disappearance peak at around 335 nm in its absorbance spectra (Figure 4A, curves b, d) and decreased at 390 nm in its FL spectra (Figure 4B, curves b, d). However, when mercapto-containing Ctl was present in the mixture solution, a typical absorption peak located at ∼335 nm and FL emission peak at 390 nm of ONCNs appeared again and increased in the UV-vis (Figure 4A, curve e) and FL (Figure 4B, curve e) spectra, respectively. The reason is that the stronger coordination/chelation interaction between the Ag^+^ ion and the mercapto groups of Ctl causes the Ag^+^ ions to fall off easily from the surface of ONCNs by forming metal chelates with mercapto-containing Ctl, inducing the FL recovery of ONCNs. Meanwhile, the UV-vis and FL spectra of ONCNs have no changes in the presence of Ctl (Figure 4A, curves b, c; Figure 4B, curves b, c), indicating that Ctl cannot affect the molecular framework of ONCNs. Therefore, these observations indicate that ONCNs-Ag^+^ nanocomposites can effectively serve as a promising platform for Ctl detection, in which ONCNs function as the fluorescent probes and Ag^+^ ions act bifunctionally as the FL quencher and Ctl recognizer.

To clarify the morphology changes of the ONCNs in different states, the TEM characterizations of ONCNs-Ag^+^ ensemble and ONCNs-Ag^+^ ensemble in the presence of Ctl drug were carried out. As shown in Figure 1, compared with the pure ONCNs, large disordered ONCNs aggregates were formed without morphology change when the Ag^+^ ion was added (Figure 1B). Subsequently, after adding Ctl drugs, the ONCN aggregates were redispersed (Figure 1C). Note that during the whole testing process, the ONCNs framework only changed its aggregation state, but the single unit of carbon nanoribbons had not changed. This result indicates that the mechanism of aggregation-induced quenching (AIQ) is the reason for the Ag^+^-induced FL quenching of ONCNs.

To further illustrate whether there were photo-induced electron transfer (PET) processes except AIQ in the quenching mechanism caused by Ag^+^ ion, the absorption and UPS spectra were carried out to measure the LUMO-HOMO energy levels of the ONCNs and ONCNs-Ag^+^ system. The conventional electron energy level calculation was usually performed using the following equations [21,35]:

E_HOMO_ = -[21.22 − (E_cutoff_ − E_onset_)] eV
(1)


E_LUMO_ = [E_HOMO_ + E_g_] eV
(2)

where the position of the disappears and rising energy counts within the UPS spectrum are defined as E_cutoff_ and E_onset_, respectively, and E_g_ is the strong absorption band within the absorption spectrum. From the UPS spectrum, the E_onset_/E_cutoff_ of the ONCNs and ONCNs-Ag^+^ complex was measured to be ~1.47/17.10 eV and ~2.92/17.30 eV, respectively (Figure 5A,B). Figure 5C shows that the E_g_ value of the ONCNs and ONCNs-Ag^+^ complex was ~4.54 and ~4.51 eV, respectively. Accordingly, based on the Equations (1) and (2), the E_HOMO_/E_LUMO_ values of the ONCNs and ONCNs-Ag^+^ complex were calculated to be about −5.59/−1.05 eV and −6.84/−2.33 eV, respectively. In other words, compared with the ONCNs E_HOMO_/E_LUMO_ values, the ONCNs-Ag^+^ ensemble E_HOMO_/E_LUMO_ values are more negative; thus, the PET process should be also responsible for Ag^+^-induced FL quenching of ONCNs (Figure 5D). To further determine this quenching process, FL lifetimes measurements in the aqueous solution were carried out, and the results are shown in Table 1 and Appendix A. A single exponential FL lifetime of ONCNs is 4.2 ns, but the global analysis of the ONCNs-Ag^+^ system decay curve gave an average lifetime of 3.7 ns with double-exponential decay. Note that the rate of electron transfer (k_ET_) and quenching efficiency (QE) for the ONCNs in the presence of Ag^+^ ion was 3.0 × 10^7^ s^−1^ and 12.1%, respectively, through analyzing the FL lifetime data, indicating quenching caused by PET only accounts for a small portion of the total quenching results (74.3%) calculated from Equation (3) [26], and the result further showed that there should be two quenching mechanisms: AIQ and PET.

QE = 1 − FL/FL_0_
(3)

where the FL and FL_0_ are the FL values of ONCNs at 390 nm in the presence or absence of Ag^+^ ion (30 μM), respectively. From the above discussions, it can be safely concluded that the AIQ and PET mechanisms caused the accumulation and electron transfer of ONCNs by Ag^+^ ions are the reasons for the FL quenching of the ONCNs. 

The combination of linear and nonlinear Stern–Volmer plots (Figure 6B) and changed lifetimes implied that Ag^+^-induced FL quenching of ONCNs obeyed the complicated FL quenching mechanism of static and dynamic synergism. In other words, the Ag^+^-induced ONCNs aggregation may create conditions for the PET process between ONCNs and ONCNs-Ag^+^ ensemble. That is, the AIQ process is accompanied by the PET process with the addition of Ag^+^ ion to ONCNs solution, resulting in high selectivity and sensitivity of ONCNs to Ag^+^ ion response finally.

From the above results and discussions, the feasibility of Ag^+^ ions detection using ONCNs as FL probes were explored under optimized conditions (50 mM Tris-HNO_3_ buffer, pH 7.0, Appendix A). From Figure 6, the FL quenching intensity of ONCN at 390 nm was proportional to the concentration of Ag^+^ ions from 0 to 30 µM (Figure 6B), showing that this probe has a high sensitivity to Ag^+^ ions. Furthermore, the quenching intensity of ONCN was nonlinear when the Ag^+^ ions concentration exceeded 30 µM (Figure 6B). These results implied that the Ag^+^-induced quenching data should follow the modified Stern–Volmer equation, combined with dynamic and static synergism quenching process, as shown in Equation (4) [31], in which k_d_ and k_s_ are the nonlinear and linear quenching constant, respectively, letter c represents Ag^+^ ion concentration, and the FL and FL_0_ are the FL values of ONCNs at 390 nm in the presence or absence of Ag^+^ ion (30 μM), respectively.
*FL*_0_/*FL* = (1 + *k_d_c*) × (1 + *k_s_c*)
(4)


From Figure 6B, this sensing platform can realize Ag^+^ ions in the linear range of 3–30 µM, with a low detection limit of ~0.78 µM (~0.043 ppb, 3σ), which was equivalent to previous Ag^+^ ions testing based on FL analysis (Appendix A). Furthermore, when 30 μM Ag^+^ ion exists, the FL of ONCNs can achieve nearly 80% quenching efficiency (Figure 6B). In addition, based on the strong affinity between Ag^+^ ions and the thiol-containing molecular compound [22], the FL intensity and state of the ONCNs-Ag^+^ ensemble may be changed by the presence of the thiol-containing Ctl drug. Therefore, to obtain the high sensitivity for Ctl drug using the ONCNs-Ag^+^ ensemble as a probe, the optimum quenching degree of 30 μM Ag^+^ ion existence was selected for Ctl drug analysis.

For Ctl drug analysis using the ONCNs-Ag^+^ ensemble system, Figure 6C displays the FL recoveries of the ONCNs-Ag^+^ ensemble with increasing concentrations of Ctl. Upon adding Ctl (30 μM) to the ONCNs-Ag^+^ ensemble, it is clearly seen that the FL ONCNs were almost completely restored, implying that this sensing platform is also sensitive to Ctl drug. Moreover, the restored process within this concentration range (<30 μM) should follow the linear Stern–Volmer equation (Figure 6D):
*FL*_0_/*FL* = *k_s_c* + 1
(5)

where *k_s_* is the linear quenching constant, letter c represents Ctl concentration, and the FL and FL_0_ are the FL values of the ONCNs-Ag^+^ system at 390 nm in the presence or absence of Ctl, respectively. The stronger binding constant between Ag^+^ ions and -SH (Ag^+^-S bond) should be responsible for FL recovery of ONCNs compared to the binding constant between Ag^+^ ion and ONCNs (Ag^+^-N bond), as a result of FL recovery caused by Ag^+^ ion falling off the surface of the ONCNs-Ag^+^ system (Figure 1). The result indicated that there was a good linear relationship between the FL recovery intensity of ONCNs and Ctl in the concentration range of 1–30.0 μM (Figure 6D), with a low detection limit of ~74 nM (n = 3, 3σ), which was equivalent to previous Ctl testing based on FL analysis (Appendix A).

Except for sensitivity, another critical index of the sensing platform is selectivity. To further study the selectivity of this sensing platform, for Ag^+^ ions analysis, we have studied the response of various metal ions to ONCNs (Appendix A), such as Mg^2+^, Na^+^, Cu^2+^, Pb^2+^, Al^3+^, K^+^, Fe^2+^, Fe^3+^, Hg^2+^, Co^2+^, Ni^2+^, Cr^3+^, Zn^2+^, Cd^2+^, Mn^2+^, Ba^2+^, Sn^2+^, and Ca^2+^ ions. All selective experiments were carried out upon the same testing conditions with an Ag^+^ ion to the metal ion concentration ratio of 3:50 except Hg^2+^ ion (30 µM Ag^+^, 50 µM Hg^2+^, and 500 µM metal ions). As shown in Appendix A and Figure 7A, except for Ag^+^ ions, other metal ions have little substantive response to ONCNs, which indicates this Ag^+^ ion sensing platform possesses excellent anti-interference for metal ions, revealing the tremendous potential of ONCNs for Ag^+^ ion detection in the real samples.

For Ctl analysis, considering the complexity of the drug detection system, possible interfering substances, including glucose, fructose, lactose, BSA, GOD, starch, sucrose, Dop, Adr, His, Ser, Tyr, Try, and PLL, were also evaluated in this constructed Ctl sensor, as illustrated in Appendix A and Figure 7B. It can be seen that after adding Ctl, the relative FL intensity of the ONCNs-Ag^+^ system was much higher. On the contrary, no FL response was observed after adding other biomolecules into the ONCNs-Ag^+^ system. These experimental results confirm the fact that ONCNs-based constructed Ctl drug sensor possesses excellent selectivity even in complex biological systems. Accordingly, to further illustrate the practicability of the sensing platform in actual sample detection, we purchased two kinds of pharmaceutical samples for feasibility analysis through the standard addition method; see Materials and Methods for the preliminary treatment and detection process. The analytical results are shown in Table 2. The RSD (<5.0%, n = 3) and Ctl recovery (94.9–107.5%, n = 3) were obtained under the testing conditions, which fully indicated that the constructed Ctl drug sensor possesses good repeatability and high accuracy, and which also implied this sensing platform fulfills the needs of real applications (Figure 7).

## 3. Materials and Methods

### 3.1. Reagents and Chemicals

Picolinic acid (PLA), captopril (Ctl), glucose, fructose, and bovine serum albumin (BSA) were supplied from Aladdin Chemical Reagent Co. Ltd. (Shanghai, China). Silver nitrate (AgNO_3_), N,N-dimethylformamide (DMF), starch, sucrose, histamine (His), glucose oxidase (GOD), serotonin (Ser), tyramine (Tyr), tryptamine (Try), phenylethylamine (PLL), lactose, dopamine (Dop), and adrenaline (Adr) were obtained from Shanghai Sinopharm Chemical Reagent Co. Ltd. (Shanghai, China). Other solvents and chemicals are of analytical grade using without further purification. Double-distilled water (DW) was used throughout the experiment.

### 3.2. Apparatus

Fluorescence (FL) and UV-vis spectra were recorded on a Jasco FP-6500 fluorescent spectrofluorometer (Jasco, Japan) and Shimadzu UV-2550 spectrophotometer (Tokyo, Japan), respectively. FT-IR spectrum was recorded on a Nicolet 5700-IR spectrometer (Waltham, MA, USA). X-ray photoelectron spectroscopy (XPS) and ultraviolet photoelectron spectroscopy (UPS) analysis were carried out on a Thermo ESCALAB 250XI X-ray photoelectron spectrometer (USA). TEM images were conducted on a JEM-2800 transmission electron microscope (JEOL Ltd., Tokyo, Japan). Fluorescence lifetime was measured using an FLS980 time-resolved spectrometer (Edinburgh, UK) with an excitation wavelength of 365 nm. 

### 3.3. Synthesis of Oxygen-Doped Nitrogen-Enrichment Carbon Nanoribbons (ONCNs) 

In brief, 0.1 g of PLA was dissolved in 8 mL DMF. After ultrasonic stirring for 1 h, the as-prepared solution of 8 mL was placed in a poly(tetrafluoro-ethylene) (Teflon)-lined autoclave and then heated at 180 °C for 16 h. After natural cooling to room temperature, the brown ONCNs solution was obtained by centrifugation at 10,000 rpm for 10 min to remove the precipitate. 

### 3.4. Optimizing the Experimental Conditions

In order to detect Ag^+^ ions and Ctl with a highly sensitive response, the pH value of the solution was optimized in our work. Firstly, 10 μL ONCNs solution (4.5 mg•mL^−1^) and 15 μL Ag^+^ ions (1 mM) solution were incubated with different pH values of 50 μL Tris-HNO_3_ buffer (50 mM) for 10 min. Then, DW was used to adjust the final volume of the mixture to 500 μL. Finally, these mixtures were left to stand for 10 min before the FL spectra were measured. Test conditions: λ_ex_ = 335 nm, and the slit widths of excitation and emission were 5 and 3 nm, respectively.

### 3.5. Fluorescent Detection Procedures for Ag^+^ and Ctl 

In a typical procedure, 10 μL ONCNs solution (4.5 mg•mL^−1^), 50 μL Tris-HNO_3_ buffer (pH 7.0, 50 mM), and the different volume of Ag^+^ ion solution (1 mM) was added into the EP tube (1.5 mL). Then, the solution was diluted with DW to the final volume of 500 μL. After thoroughly shaking the mixture at room temperature, the fluorescence spectra were measured by using the micro quartz cuvette (500 μL). Likewise, 10 μL ONCNs (4.5 mg•mL^−1^) solution, 50 μL Tris-HNO_3_ buffer (pH 7.0, 50 mM), and 15 μL Ag^+^ ions (1 mM) were added into the EP tube (1.5 mL) in order to form the probe of metal complexes (ONCNs-Ag^+^). Then, different concentrations of Ctl were added, and the mixture with the final volume of 500 μL diluted by DW was shaken thoroughly at room temperature. Test conditions as described above.

### 3.6. Selectivity Investigation for Ctl

In the selective experiment, 10 μL ONCNs (4.5 mg•mL^−1^) solution and 15 μL Ag^+^ ions (1 mM) were incubated for 10 min in Tris-HNO_3_ buffer (pH 7.0, 50 mM). Then, a series of competitive molecules, such as glucose, lactose, fructose, sucrose, GOD, BSA, starch, Dop, His, Ser, Adr, Tyr, Try, and PLL, were mixed with that solution and incubated for another 5 min at room temperature. The final volume of 500 μL was adjusted with DW, and the mixture was equilibrated for 10 min at room temperature before recording the spectrum measurements. The concentration of Ctl was 30 μM; the concentration of Dop and Ser was 50 μM, respectively; and other interfering molecules were all 500 μM. Test conditions as described above.

### 3.7. Procedure for Pharmaceutical Sample 

A pharmaceutical sample, labelled as 25 mg Ctl per tablet, was purchased from Shanghai pharmaceutical Co., Ltd. (Shanghai, China). Three tablets were ground to a homogeneous powder. The powder containing 15.5 mg Ctl was weighed and dissolved in DW to form a solution of 100 μM. The mixture was filtered with a 0.45 μm membrane filter after ultrasonication for 30 min. Then, the supernatant was collected by centrifuging at 10,000 *g* for 10 min and stored at 4 °C in the dark for further use. Standard procedures were followed, then the supernatant was diluted and detected. In order to evaluate the accuracy of the present “on-off-on” sensor, the Ctl standard solutions were added to the diluted sample solutions, and the recovery tests were carried out. For analysis of Ctl samples, 10 μL ONCNs solution (4.5 mg•mL^−1^) and 15 μL Ag^+^ ions (1 mM) were incubated for several seconds in Tris-HNO_3_ buffer (pH 7.0, 50 mM). Then, different concentrations of Ctl solution were mixed with this solution and incubated for 10 min at room temperature. The final volume of 500 μL was adjusted with DW, and the mixtures were equilibrated for 10 min at room temperature before recording the FL spectrum measurements. Test conditions as described above. All measurements were repeated three times.

## 4. Conclusions

In summary, a simple FL sensor was successfully developed for highly sensitive and selective detection of Ctl based on Ag^+^-induced quenching of the self-assembly ONCNs. The fluorescent intensity of ONCNs is decreased in the presence of Ag^+^, whereas the quenching fluorescent of ONCNs is increased with the addition of Ctl to the ONCNs-Ag^+^ metal system. The Ag^+^-induced quenching mechanism has been carefully studied and discussed in detail experimentally. That is, based on the mechanism of synergism of PET and AIQ, the ONCNs were used as an effective fluorescent probe for the measurement of Ag^+^ and Ctl with good properties of convenience, rapidity, excellent linear relationships, high sensitivity, and selectivity. What is more, the potential in practical applications of our constructed Ctl drug sensing platform has been proved by detecting Ctl in commercial tablet samples, which provides a new application for these carbon-based nanomaterials.

## Data Availability

Data are contained within the article.

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
