# Peer review of "Synergism of Photo-Induced Electron Transfer and Aggregation-Induced Quenching Mechanisms for Highly Sensitive Detection of Silver Ion and Captopril"

_molecules, 2023, doi:10.3390/molecules28093650_

Round 1

Reviewer 1 Report

I read the manuscript entitled “Coordination of photoelectron transfer and aggregation-induced quenching mechanisms for highly sensitive detection of silver ions and captopril

” and have the following suggestions and comments to improve the state of the study.

The text on lines 120,121 looks very similar to the text on lines 124, 125, check and correct.

If the final volume was 500 μL what kind of cuvette was used for the measurements (should be mentioned)?

On lines 124-125 why do Ag+ ions appear if we determine Ctl?

Where are Tables S1-S2 and Figures S1-S5, in the MDPI platform the Supplementary Materials file is not loaded.

Minor points: line 90: N,N-Dimethyl formamide (DMF),= N,N-dimethylformamide (DMF),  line 90: were optimized = was optimized ; line 307: obeyed the complicatedly = obeyed the complicated; line 323: was explored =were explored

Author Response

Comments and Suggestions for Authors

Reviewer #1: I read the manuscript entitled “Coordination of photoelectron transfer and aggregation-induced quenching mechanisms for highly sensitive detection of silver ions and captopril” and have the following suggestions and comments to improve the state of the study.

We really appreciate for your time reading our paper and evaluation about our work. The comments are valuable and very helpful for revising and improving the quality of our paper. We have studied comments carefully and have made correction which we hope meet with approval.

The text on lines 120,121 looks very similar to the text on lines 124, 125, check and correct.

Answer: Thanks for the reviewer’s valuable suggestion. According to reviewer’s recommendation, the relevant statements have been re-written in the section of "2.5. Fluorescent detection procedures for Ag+ and Ctl". (Please see the revised manuscript)

If the final volume was 500 μL what kind of cuvette was used for the measurements (should be mentioned)?

Answer: Thanks for the reviewer’s question. The micro quartz cuvette with a volume of 500 μL was used for the fluorescence measurements in our experiments, and the relevant statement have been supplemented in the section of "2.5. Fluorescent detection procedures for Ag+ and Ctl". (Please see the revised manuscript)

On lines 124-125 why do Ag+ ions appear if we determine Ctl?

Answer: Thanks for the reviewer’s question. In fact, our experiment results display an interesting "on-off-on" three-state emission with the stepwise addition of Ag+ ion and Ctl. That is, the fluorescence intensity of ONCNs firstly quenched in the presence of Ag+ ion, and then the fluorescence intensity of ONCNs can be restored after adding Ctl to the above mixture. Hence, the ONCNs-Ag+ complex can be used as “turn-on” fluorescence probe for sensing Ctl due to the formation of a stronger Ag-S bond between Ag+ and Ctl, which results in the detachment of Ag+ ion from the surface of ONCNs, thereby restoring the fluorescence of ONCNs. In other words, Ag+ ion is a key bridge for ONCNs as fluorescent probe for detecting Ctl in this work. Meanwhile, the relevant statements have been provided in the last paragraph of the "Introduction". (Please see the revised manuscript)

Where are Tables S1-S2 and Figures S1-S5, in the MDPI platform the Supplementary Materials file is not loaded.

Answer: Thanks for the reviewer’s kindly reminding, the Supplementary Materials file has been re-uploaded on the MDPI platform.

Minor points: line 90: N,N-Dimethyl formamide (DMF) = N,N-dimethylformamide (DMF), line 90: were optimized = was optimized; line 307: obeyed the complicatedly = obeyed the complicated; line 323: was explored =were explored

Answer: Thanks for the reviewer’s kindly comment and suggestion on our manuscript. According to reviewer’s recommendation, the relevant statements have been amended in the revised manuscript. (Please see the revised manuscript).

Reviewer 2 Report

The above referenced manuscript presented the synthesis and characterization of luminescent oxygen-doped nitrogen-enrichment carbon nanoribbons (ONCNs) toward silver ions and captopril detection. The experimental observation on the luminescence evolution of carbon nanoribbons seems to be interesting when adding Ag+ and/or then captopril molecules into the solution. The manuscript, however, was not well written and organized for clarity and accuracy; these render the science presented unclear and not easy reading, especially in the section of Results and Discussion. Some statements or description were incorrect. I did not recommend it for publication in the present form in Molecules. The following was some specific comments.

1.   For the title, why to call photoelectron transfer, since the irradiation did not eject the electrons from the surface of nanoribbons? The “coordination” should read “synergism”?

2. For the sake of clarity, the authors are suggested to logically illustrate the experimental results obtained from each figure one by one, and to avoid repeatedly presenting the same data in different paragraphs or mixing the results of different figure in one paragraph. For example, the results of Figure 2A were presented two times in page 4 and 5.

3.  One TEM on single nanoribbon or an enlarged TEM image is necessary to check these nanoribbons in some detail. Please give the scale bar for each TEM image clearly.

4. Please check the assignment of absorption peak at 265 nm to σπ electron transfer, it is incorrect by considering the molecular orbital theory.

5.  How did the authors get the data in Figure 5C? Are they those in Figure 4A? To evaluate the band-gap energy by extrapolating the linear part of absorbance, the tick label on the Y axis, especially the zero value should be given. In addition, the linear part should correspond to the steep absorption edge near 300 nm in Figure 4A, rather than the weaker part. The extrapolation done by the authors seems to be somewhat arbitrary.

6.  Does Ag+ have absorption in UV region? If so, it will affect the following spectral analysis.

7.  How did the authors determine the cut-off and initial binding energies of the UPS spectrum, please comment on that in the text.

8.  Could the authors provide any experimental evidence, such as FTIR and XPS, to demonstrate the binding of Ag+ with nitrogen containing moieties at the surface of ONCNs, and to quantitatively evaluate the amount of Ag+ on the nanoribbon? In addition, since Ag+ ions are only located on the surface of sizeable nanoribbion, this would not be expected to greatly change the energy levels of LUMO and HOMO, and then the energy gap between them.  

9. If electron transfer induced the luminescence quenching, the following decay should commonly be nonexponential rather than single exponential again.

10.   It is not reasonable to have a significant difference in the quenching efficiency (QE) respectively calculated by FL and lifetime methods.

11.  The equation 4 and the following one (eq. 5?) seem to be incorrect.

12. Aggregation indeed occurred in solution, but how to know it induced the luminescence quenching greatly. So for dry ONCNs powders with serious aggregation in that case, did the quenching occur? 

Author Response

Comments and Suggestions for Authors

Reviewer #2: The above referenced manuscript presented the synthesis and characterization of luminescent oxygen-doped nitrogen-enrichment carbon nanoribbons (ONCNs) toward silver ions and captopril detection. The experimental observation on the luminescence evolution of carbon nanoribbons seems to be interesting when adding Ag+ and/or then captopril molecules into the solution. The manuscript, however, was not well written and organized for clarity and accuracy; these render the science presented unclear and not easy reading, especially in the section of Results and Discussion. Some statements or description were incorrect. I did not recommend it for publication in the present form in Molecules. The following was some specific comments.

We really appreciate for your time reading our paper and evaluation about our work. The comments are valuable and very helpful for revising and improving the quality of our paper. We have studied comments carefully and have made correction which we hope meet with approval.

  1. For the title, why to call photoelectron transfer, since the irradiation did not eject the electrons from the surface of nanoribbons? The “coordination” should read “synergism”?

Answer: Thanks for the reviewer’s valuable suggestion. We agree with the reviewers’ point of view. According to the reviewer’s suggestion, the statements of "photoelectron transfer" and “coordination” have been corrected into "photo-induced electron transfer" and “synergism”, respectively. (Please see the revised manuscript)

  1. For the sake of clarity, the authors are suggested to logically illustrate the experimental results obtained from each figure one by one, and to avoid repeatedly presenting the same data in different paragraphs or mixing the results of different figure in one paragraph. For example, the results of Figure 2A were presented two times in page 4 and 5.

Answer: Thanks for the reviewer’s valuable suggestion. According to the reviewer’s suggestion, the analysis of the experimental results has been re-organized and re-written in the revised version. (Please see the section “3. Results and discussion” in the revised version)

  1. One TEM on single nanoribbon or an enlarged TEM image is necessary to check these nanoribbons in some detail. Please give the scale bar for each TEM image clearly.

Answer: Thanks for the reviewer’s valuable suggestion. The enlarged TEM image has been provided in the Figure S1 (Please see the revised Supplementary Materials file). And the clear scale bar for each TEM image have been given in Figure 1 and Figure S1 (Please see the revised manuscript and Supplementary Materials file).

  1. Please check the assignment of absorption peak at 256 nm to σ→π electron transfer, it is incorrect by considering the molecular orbital theory.

Answer: We appreciate the reviewer for the valuable suggestion. According to your suggestions, we have consulted and carefully surveyed a large number of literatures to further ascertain the attribution of absorption peak at 256 nm, and the investigation results show that the absorption peaks at ~240--260 nm of carbon-based nanomaterials, including doped-graphene and -carbon nanotube, mostly belong to σ → π transition of the aromatic sp2 domains (Adv. Mater. 2011, 23, 776-780; Adv. Funct. Mater., 2014, 24, 3021-3026; Chem. Eur. J. 2017, 23, 665-675; J. Mater. Chem. B, 2018, 6, 1771-1781; J. Mater. Chem. B, 2023, 11, 1523-1532; Adv. Sci., 2023, 10, 220762).

  1. How did the authors get the data in Figure 5C? Are they those in Figure 4A? To evaluate the band-gap energy by extrapolating the linear part of absorbance, the tick label on the Y axis, especially the zero value should be given. In addition, the linear part should correspond to the steep absorption edge near 300 nm in Figure 4A, rather than the weaker part. The extrapolation done by the authors seems to be somewhat arbitrary.

Answer: Many thanks for the reviewer’s valuable advice. As you said, the data of Figure 4A and Figure 5 were obtained through UV-vis testing, in which Figure 4A was obtained from the UV-vis spectrum data directly. Here, we only need to know the change in the position of the characteristic absorption peak, so zero-setting was not necessary. However, we want to obtain the absorption edge of Eg from the UV-vis absorption spectrum, therefore, zero-setting when testing the UV-vis data in Figure 5C was necessary, and processed according to the formula (Energy = 1240/λnm, Ref. Langmuir 2021, 37, 949-956). Meanwhile, according to the reviewer’s  suggestions, the tick label on the Y axis was also been added in Figure 5C (Please see the Figure 5C in the revised version).

In the calculation formula of the material energy level of the lowest energy unoccupied molecular orbital (LUMO)-highest energy occupied molecular orbital (HOMO) energy level, the conventional LUMO-HOMO energy level calculation was performed by the following equations (Chem. Rev., 1998, 98, 1089-1107; Electrochim Acta, 2013, 96, 13-17; Langmuir 2021, 37, 949-956; J. Mater. Chem. B, 2023, 11, 1523-1532):

EHOMO = -[21.22 - (Ecutoff - Eonset)] eV               (1)

ELUMO = [EHOMO + Eg] eV                                 (2)

where, the location of the onset and cutoff binding energies in the UPS spectrum are defined as the Eonset and Ecutoff, respectively, and Eg is the absorption edge in the UV-vis absorption spectrum. That is, here, it is necessary to obtain the absorption edge energy of Eg rather than the excitation or emission position Eg energy (excitation/emission energy at steep absorption edge of ~300 nm or 390 nm), so the steep absorption edge near 300 nm does not need to be calculated.

  1. Does Ag+ have absorption in UV region? If so, it will affect the following spectral analysis.

Answer: Thanks for the reviewer’s good question. According to your suggestions, the UV-vis absorption spectrum of Ag+ ion was carried out (Figure R1, black line). According to the results of our experiments, Ag+ ion have no absorption in UV region, therefore, Ag+ ion will not affect the following spectral analysis in our work.

Figure R1 The UV-vis spectra of Ag+ ion (black line), Ctl (red line) and ONCNs (green line).

  1. How did the authors determine the cut-off and initial binding energies of the UPS spectrum, please comment on that in the text.

Answer: According to the reviewer’s suggestions, the Ecutoff and Eonset within the UPS spectrum have been redefined on that in the revised text. (Please see the revised manuscript)

  1. Could the authors provide any experimental evidence, such as FTIR and XPS, to demonstrate the binding of Ag+ with nitrogen containing moieties at the surface of ONCNs, and to quantitatively evaluate the amount of Ag+ on the nanoribbon? In addition, since Ag+ ions are only located on the surface of sizeable nanoribbion, this would not be expected to greatly change the energy levels of LUMO and HOMO, and then the energy gap between them.

Answer: According to your suggestions, the XPS survey spectrum and high-resolution N1s spectrum of ONCNs in the presence of Ag+ ions were provided (Figure R2). From the results of the XPS (Figure R2A) and high-resolution N1s (Figure R2B) spectra, the interaction between Ag+ ion and ONCNs is indeed through strong Ag+-N bond, and the results of the elemental analysis from the XPS data indicate that the amount of Ag+ ion on the surface of ONCNs was evaluated about 11.87 At%.

Figure R2 A) The XPS survey spectra of ONCNs (curve a) and ONCNs-Ag+ metal complexes (curve b); B) The high-resolution N1s spectrum of ONCNs-Ag+ metal complexes.

From the Figure 5, the experimental results show that the energy levels of LUMO and HOMO of ONCNs-Ag+ metal complexes become more negative, which might be due to the fact that Ag+ ions bind these ONCNs together, shortening the distance between each other, thus resulting in undergoing π···π conjugation between each other or rearrangement of electronic configurations of ONCNs-Ag+ metal complexes, and their energy levels including energy gap became smaller ultimately.

  1. If electron transfer induced the luminescence quenching, the following decay should commonly be nonexponential rather than single exponential again.

Answer: Many thanks for the reviewer’s valuable advice. As you said, if electron transfer induced the luminescence quenching, the following decay should commonly be nonexponential rather than single exponential again. However, from the experimental analysis results, we know that the quenching of ONCNs in the presence of Ag+ ion is mainly caused by two quenching mechanisms: the aggregation-induced quenching and photo-induced electron transfer mechanisms. The aggregation-induced quenching mechanism is the main factor leading to the quenching of ONCNs in the presence of Ag+ ions, while the quenching induced by photo-induced electron transfer mechanism may play a minor role. Therefore, the factor of photo-induced electron transfer mechanism might be not well reflected in the luminescent lifetime experiment.

  1. It is not reasonable to have a significant difference in the quenching efficiency (QE) respectively calculated by FL and lifetime methods.

Answer: We appreciate the reviewer for the valuable suggestion. In fact, if the quenching of the probe is entirely caused by the photo-induced electron transfer mechanism, the fluorescence quenching efficiency (QE) obtained by FL and lifetime methods according to the reported literature (Langmuir 2021, 37, 949-956) is almost identical to the results obtained by our fluorescence quenching experiments previously (Dyes and Pigments 2022, 208, 110859; J. Mater. Chem. B, 2023, 11, 1523-1532). It is noteworthy that the quenching of ONCNs in our system is caused by two quenching mechanisms: the aggregation-induced quenching and photo-induced electron transfer mechanisms. Where, the photo-induced electron transfer mechanism plays a small role for the quenching of ONCNs in the presence of Ag+ ion. Therefore, the quenching efficiency obtained by FL and lifetime methods in this system should account for a small part of the total quenching efficiency of the ONCNs in the presence of Ag+ ion, which is consistent with our calculation results.

  1. The equation 4 and the following one (eq. 5?) seem to be incorrect.

Answer: Thanks for the reviewer’s good question. We have consulted and carefully surveyed the reported literature in order to further ascertain the validity of eqs. 4 and 5. According to this literature (Microchim. Acta 2022, 189, 241), we know that when the SV curve is curved, it indicates that static and dynamic quenching has occurred. In this case, a modified Stern-Volmer eq. should be used: Iₒ/I = (1 + Kd[quencher])⋅(1 + Ks[quencher]), where Kd is the dynamic quenching constant, and Ks is the static quenching constant; and when the SV plot is linear, it indicates that static quenching has occurred. In this case, a type Stern-Volmer plot of (Iₒ/If) = 1 + Ks[quencher] should be used, where Ks is the static quenching constant. And from the fluorescence quenching curve of ONCNs in the presence of Ag+ ions, the fluorescence quenching process follows the modified Stern-Volmer equation via static and dynamic synergism quenching mechanism: FL/FL0 = (1+kdc)×(1+ksc)                  (4); and from the fluorescence recovery curve of ONCNs-Ag+ metal complexes in the presence of Ctl, restored process should follow the linear Stern-Volmer equation of the static quenching mechanism: FL/FL0 = ksc + 1                   (5). And some errors in this equation have also been corrected in the revised version. (Please see the revised manuscript)

  1. Aggregation indeed occurred in solution, but how to know it induced the luminescence quenching greatly. So for dry ONCNs powders with serious aggregation in that case, did the quenching occur?

Answer: Thanks for the reviewer’s good question. According to the results of our experiments, the ONCNs were well dispersed in solution before adding Ag+ ions, while aggregation indeed occurred after adding Ag+ ions (please see the SEM images in the revised manuscript). Combined with the change of fluorescence signal, we conclude that the fluorescence quenching is mainly based on aggregation induction. The relevant statements have been provided in page 7: "Note that during the whole testing process, the ONCNs framework only changed its aggregation state, but the single unit of carbon nanoribbons had not changed. This result indicates that the mechanism of aggregation-induced quenching (AIQ) is the mainly reason for the Ag+-induced FL quenching of ONCNs." (please see the SEM images in the revised manuscript)

Other improvements of the original manuscript please see the revised manuscript. These changes will not influence the content and framework of the paper. Once again, thank you very much for your comments and suggestions.

Round 2

Reviewer 1 Report

The authors have considered my comments. This paper could be accepted for publication. 

Author Response

Responses to Referees’ Comments

Thank you very much for your recognition of our work.

Reviewer 2 Report

Thanks to the author’s revision following some of my comments. However, I still did not agree with some of their statement/conclusions.

1.      Most of the nanoribbons are expected to bind with the Ag+ after adding Ag+ into the solution, then if the electron transfer occurred indeed, it should occur between the luminescent species of nanoribbons and Ag+, rather than among the different nanoribbons, which was schematically shown in Figure 5d.

2.      For comment No. 5, I did not agree with the methods, they adopted, of determining the Eg by extrapolating the linear part of weak absorbance, which commonly arose from surface state or others. In addition, the absorption of ONCNs+Ag+ was also not shown until its onset appeared. The values are incorrect.

3.      For comment No. 7, particularly one could not easily distinguish the onset of the rising energy counts from the UPS spectra, so the values of 1.47 and 2.92 eV they gave are arbitrary.

4.      For comments No. 9 and 10, if electron-transfer indeed occurred during luminescence quenching, even with a small part, commonly we shall observe the nonexponential decay in the initial stage. In addition, if both FL and lifetime methods only account for a part of quenching, then how many of the total quenching efficiency? In my opinion, the FL method is calculating the total quenching.

5.      The aggregation induced quenching does not mean a static quenching. It could also produce the nonexponential fluorescence decay.

6.      For comment No. 11, the correct expression for the equations 4 and 5 should be FL0/FL= (1+kdc)×(1+ksc) and FL0/FL= 1+ksC. In addition, why the dominant quenching mechanisms are different in the processes of adding Ag+ and adding Ctl (recovery). It seems to be contradictory. Commonly, they should be same.

7.      The comment No.12 was not clearly answered. “So for dry ONCNs powders with serious aggregation in that case, did the quenching occur? In addition, since the as-prepared nanoribbons are longer in length, so are they stable in the aqueous solution or will precipitate from the solution after long term of storage? If they separated from the solution, did they luminesce? These phenomena are favorable in identifying the quenching mechanism.

8.      Line 220-221, page 6, why H+ or OH- could quench or enhance the PL? Please comment on that.

9.      Did the authors measure the luminescence and fluorescence decay in the aqueous solution? Please comment on that in the text.

10.  In Table 1, it is enough to correct the number of τ to one decimal places. The calculated transfer rate is so small, it is impossible for a real process.

11.  Line 159-160, page 4, the description on the size is not clear. Lines 199-201, page 5; Lines 206-209, lines 212-213, lines 235-236, lines 239-240,page 6; please alter for clarity.

12.  Lines 250-251, curves a, c should read curves b, c.
